



# Vehicular ammonia emissions: An underappreciated emission source in densely-populated areas

Yifan Wen[1], Shaojun Zhang[1, 2, 3*], Ye Wu[1, 2, 3], Jiming Hao[1, 2, 3]

[1]School of Environment, State Key Joint Laboratory of Environment Simulation and Pollution Control, Tsinghua University, Beijing, 100084, China
[2]State Environmental Protection Key Laboratory of Sources and Control of Air Pollution Complex, Beijing, 100084, China
[3]Beijing Laboratory of Environmental Frontier Technologies, School of Environment, Tsinghua University, Beijing 100084, China

*Correspondence to*: Shaojun Zhang (zhsjun@tsinghua.edu.cn)

**Abstract.** On-road ammonia ($NH_3$) emissions play a significant role in fine particulate matter ($PM_{2.5}$) formation in urban areas, posing severer risks for human health. Limited studies have depicted the spatial and temporal variations of on-road $NH_3$ emissions, in particular lacking detailed quantification of their contributions within densely-populated areas. In this study, we established a comprehensive vehicular $NH_3$ emission model and compiled a gridded on-road $NH_3$ emission inventory with high spatial (3 km × 3 km), and temporal (monthly) resolutions for mainland China. China's annual vehicular $NH_3$ emissions are estimated to increase from 32.8 kt to 87.1 kt during the period of 2000-2019. Vehicular $NH_3$ emissions are significantly concentrated in densely-populated areas where agricultural emissions have relatively lower intensity. It is found that vehicular $NH_3$ emissions could exceed agricultural emissions in the grids containing 23.0% of the Chinese population in 2019 (approximately 326.6 million people), and this ratio is up to 29.4% in winter. For extreme populous megacities such as Beijing and Shanghai, vehicular $NH_3$ emissions exceed agricultural emissions where 69.2% and 72.0% of population resides, respectively. Thus, the significant role of on-road $NH_3$ emissions in populated areas may have been underappreciated. This study gave a better insight into the absolute value and relative importance of on-road $NH_3$ emissions in different regions, seasons and population densities in China, which is important in terms of the air quality implications.

## 1 Introduction

As the leading alkaline gas and one of the major reactive nitrogen species in the atmosphere, ammonia ($NH_3$) plays a vital role in fine particulate matter ($PM_{2.5}$) pollution and nitrogen deposition. $NH_3$ readily neutralizes with acidic species from sulfur dioxide ($SO_2$) and nitrogen oxides ($NO_X$) precursors to form secondary organic aerosols (SOA) (Lv et al., 2022; Chu et al., 2016), which not only enhances regional haze but also threatens public health (Huang et al., 2014; Ru-Jin et al., 2014; Pan et al., 2016). It is found that $NH_3$ emissions contributed larger to $PM_{2.5}$ than NOx emission globally and in most countries, indicating that $PM_{2.5}$ is more strongly $NH_3$-limited than $NO_X$-limited (Gu et al., 2021). There are increasing evidences indicating that the reduction of $NH_3$ emissions should be more efficient than other particle precursors at mitigating haze



pollution (Fu et al., 2017; Gu et al., 2021), highlighting the priority for ammonia regulation. After removal from atmosphere, $NH_3$ and ammonium ($NH_4^+$) from both wet and dry deposition may also contribute to soil acidification, eutrophication, and even to a reduction of biodiversity (Stevens et al., 2004; Li et al., 2016). Therefore, efforts to better understand and control $NH_3$ emissions are essential.

Although agriculture dominates the total anthropogenic $NH_3$ emissions at global scales (Paulot et al., 2014), increasing studies have pointed out the significant role of on-road $NH_3$ emissions in urban areas (Chang et al., 2016; Farren et al., 2020; Fenn et al., 2018; Sun et al., 2017b). On-road $NH_3$ emissions are highly concentrated in densely-populated areas where agricultural emissions rarely exist (Sun et al., 2017a). It is reported that on-road $NH_3$ emissions have exceeded agricultural emissions where nearly half the U.S. population resides (Sun et al., 2017a; Fenn et al., 2018). Pronounced bimodal diurnal variations in $NH_3$
concentration consistent with traffic patterns were also observed in many megacities, suggesting a significant contribution of on-road $NH_3$ emissions in urban areas (Wang et al., 2015; Pandolfi et al., 2012). What's more, on-road $NH_3$ emissions are co-emitted with $NO_X$ in dense, highly urbanized areas, and may have a more effective pathway to particle formation than agricultural $NH_3$ emitted in rural, low-$NO_X$ areas (Farren et al., 2020). Thus, on-road $NH_3$ emissions could be critical for public health in urban areas due to their contribution to $PM_{2.5}$ formations, since more than half of the global populations live within
cities (World Bank Group, 2022).

There are two major sources for vehicular $NH_3$ emissions - gasoline vehicles equipped with three-way catalysts (TWC) and diesel vehicles equipped with selective catalytic reduction (SCR). $NH_3$ is the by-product from the reduction of nitric oxide (NO) for gasoline vehicles equipped with TWC (Livingston et al., 2009; Bishop and Stedman, 2015). Also, $NH_3$ leakage during the injection of urea to SCR system, commonly termed "ammonia slip", is gaining importance with the extensive applications
of SCR in diesel vehicles (Suarez-Bertoa et al., 2017; Mendoza-Villafuerte et al., 2017; He et al., 2020). With the extensive equipment of TWCs and SCR for the latest emission standards, $NH_3$ contributes increasing fractions of the reactive nitrogen species emitted by vehicles in the recent decade (Bishop et al., 2010; Sun et al., 2017b; Fenn et al., 2018). However, regulations for on-road $NH_3$ emissions are far behind other traffic-related pollutants (i.e., $NO_X$, PM, CO and HC) (Wu et al., 2017b). Currently, the heavy-duty Euro/China VI is the only emission standard legislates an $NH_3$ emission cap (10 ppm as the cycle-
average slip limit) aiming at restrain SCR slip (Sun et al., 2017a). To response the increasing concern regarding vehicular $NH_3$ emissions, stringent limits of 20 mg/km for light-duty vehicles and 65 mg/kWh for heavy-duty vehicles have been introduced in a proposal version of future Euro 7/VII regulations (European Commission, 2022). The introduction of $NH_3$ emission limits the installation will require installation of specific after-treatment devices; for example, ammonia slip catalysts (ASC) and Clean Up Catalyst (CUC) are expected to ensure Euro 7/VII vehicles to comply with these proposed limits (Torp et al., 2021).
$NH_3$ emission inventories can significantly affect the accuracy of $PM_{2.5}$ modeling and play a crucial role in the refinement of mitigation strategies. Numerous studies have established $NH_3$ emission inventories on global level (Meng et al., 2017), national level (Fenn et al., 2018; Xing et al., 2013; Kang et al., 2016; Li et al., 2021), and regional level (Zhao et al., 2012; Zheng et al., 2012). However, these studies failed to take into account the spatial distribution of on-road $NH_3$ emissions and the potential for relatively higher emissions from mobile sources in dense, highly urbanized areas. Also, $NH_3$ emissions from the



transportation sector are thought to be highly underestimated in global (Meng et al., 2017), the US (Sun et al., 2017b) and UK (Farren et al., 2020), mainly due to the large uncertainties remain in vehicular $NH_3$ emission factors (EF) and traffic activity data (Meng et al., 2017). The exact contribution of traffic sources to $NH_3$ emissions in various spatial scales is still an area of debate, especially in densely-populated areas. Therefore, comprehensive vehicular $NH_3$ EFs and high quality on-road $NH_3$ emission inventories are urgently required for air quality modelling and future $NH_3$ regulations.

In this study, we established a comprehensive vehicular $NH_3$ emission factor (EF, unit in mg/km) model including both gasoline and diesel vehicles. The long-term trend of vehicular $NH_3$ emissions from 2000 to 2019 was estimated based on the EF factors and province-level traffic activity data. Then a highly-resolved vehicular $NH_3$ emission inventory with high spatial (3 km×3 km) and temporal (monthly) resolutions was compiled for mainland China, and the relative contribution of on-road vehicles to total anthropogenic $NH_3$ emissions were analyzed among different seasons and population densities. This study is

aimed to give a better insight into the absolute value and relative importance of on-road $NH_3$ emissions in different regions, seasons and population density, which is important in terms of the air quality implications.

## 2 Methodology and data

### 2.1 Establishment of vehicular $NH_3$ emission factor model

We developed a comprehensive vehicular $NH_3$ EF model for both gasoline and diesel vehicles based on the local exhaust

measurement data in China. Nine vehicle categories and six emission standards (i.e., China 1/I to China 6/VI) were divided according to the official vehicle registration rules in China (see Table S1).

For gasoline vehicles, Huang et al (Huang et al., 2018) revealed the strong correlation between $NH_3$ emissions and modified combustion efficiency (MCE), an indicator calculated based on CO and $CO_2$ emissions. In this study, CO and $CO_2$ EFs for gasoline vehicles were obtained from EMBEV model, the archetype model for China's National Emission Inventory

Guidebook (Zhang et al., 2014). Several studies have found significant temperature-dependence for $NH_3$ emissions from light-duty gasoline vehicles (LDGVs) that increased as the temperature decreased, mainly linked to rich combustion during cold-start operations (Selleri et al., 2022; Suarez-Bertoa et al., 2017). The latest version of EMBEV updated the cold-start sub-module and developed comprehensive ambient temperature corrections that can characterize the spatial and monthly variations in EFs across China (Wen et al., 2021), enable us to estimate $NH_3$ EFs under various seasons and provinces.

For diesel vehicles, we obtained $NH_3$ measurement data from a fleet of heavy-duty diesel vehicles (HDDVs) (China III to China V) using PEMS and dynamometer (He et al., 2020). As China VI has not been widely implemented until 2020, the $NH_3$ EFs for HDDVs were categorized into pre-China IV (without SCR) and China IV/V (majorly equipped with SCR). The test results indicated that the introduction of SCR systems to diesel fleets might risk higher $NH_3$ emissions, though $NH_3$ emissions varied significantly among tested HDDVs. $NH_3$ EFs of other diesel vehicles were calculated based on the relative fuel

consumptions comparing with HDDVs. We did not introduce temperature corrections due to the lack of measurements.



## 2.2 Bottom-up estimation of long-term vehicular $NH_3$ emissions

In this study, vehicular $NH_3$ emissions inventories by province and month were calculated for mainland China from 2000 to 2019 based on a bottom-up method, involving vehicle population, vehicle kilometer traveled (VKT), and $NH_3$ EFs by province, calendar year, month and vehicle category, as Eq. (1) illustrates.

$$E_{p,y,m} = \sum_v \sum_f \sum_{es} VP_{v,f,es,p,y} \times VKT_{v,f,es,p,y} \times EF_{v,f,es,p,m} \times 10^{-9} \tag{1}$$

where $E_{p,y,m}$ is the monthly vehicular $NH_3$ emissions of province $p$ in calendar year $y$ from 2000 to 2019, units in t; $VP_{v,f,es,p,y}$ is the vehicle population of province $p$ in calendar year $y$, defined by vehicle category $v$, fuel type $f$, and emission standard $es$, units in veh; $VKT_{v,f,es,p,y}$ is the corresponding annual-average VKT, units in km/year; $EF_{v,f,es,p,m}$ is the $NH_3$ EFs in province $p$ and month $m$, defined by vehicle category $v$, fuel type $f$, and emission standard $es$, units in mg/km. The province-level vehicle

populations were obtained from National Bureau of Statistics of China (NBSC) and further processed to match up with the resolution and scale of this study (method reported by Wu et al. (Wu et al., 2016)). The annual-average VKT for various vehicle categories was estimated based on previous survey results regarding vehicle usage in China (Wu et al., 2016; Wu et al., 2017a; Zhang et al., 2014). The $NH_3$ EFs by province, month and vehicle category are obtained from the vehicular $NH_3$ emission factor model established in 2.1.

To validate the accuracy of bottom-up estimations, we compared the $NH_3$ emissions from gasoline vehicles with the top-down estimation based on annual gasoline consumption and fuel-based EFs from related studies, as Eq. (2) illustrates.

$$E_{top-down,y} = FC_y \times 0.85 \times \frac{M(CO_2)}{M(C)} \times EF_y\left(NH_3/CO_2\right) \times \frac{M(NH_3)}{M(CO_2)} \tag{2}$$

where, $E_{top-down,y}$ is the top-down estimation of $NH_3$ emissions from gasoline vehicles in calendar year $y$, units in t; $FC_y$ is the annual gasoline consumption in calendar year $y$, units in t; $FC_y$ is then converted to $CO_2$ emissions based on the carbon content

(0.85 g/kg) of gasoline and the molar mass ratio of $CO_2$ and carbon; $EF_y\left(NH_3/CO_2\right)$ is the fuel-based $NH_3$ EFs of gasoline fleet in China in calendar year $y$ from Sun et al (Sun et al., 2017b), units in ppbv/ppmv $CO_2$; $M(CO_2)$, $M(C)$ and $M(NH_3)$ are the molar masses of $CO_2$, carbon and $NH_3$, respectively, units in g/mol. The annual gasoline consumption data were obtained from National Bureau of Statistics of China (NBSC).

## 2.3 Compilation of the gridded $NH_3$ emission inventories

$NH_3$ emission data from other anthropogenic sources by province were obtained from the updating works of Zheng et al (Zheng et al., 2019). The agricultural and vehicular $NH_3$ emissions were compiled at 3 km×3 km and monthly resolutions for mainland China in 2019. Monthly variations in agricultural emissions referred to Zhang et al (Zhang et al., 2018). Emissions from fertilizer applications and livestock productions were presented at the provincial level first and then allocated by grassland



areas and rural residential areas, referring to Li et al (Li et al., 2021). The land cover data with a resolution of 1 km was
obtained from China's National Land Use and Cover Change (CNLUCC) dataset (Xu et al., 2018).

For on-road NH$_3$ emissions, we allocated the total vehicular NH$_3$ emissions of each province to 3 km×3 km grids based on the
relative ratio of traffic indicator in each grid (see Eq. (3) and (4)).

$$E_{p,i} = \frac{R_{traffic,i}}{\sum_1^n R_{traffic,i}} \times E_p \tag{3}$$

$$R_{traffic,i} = \left(a \cdot L_{rank1,i} + b \cdot L_{rank2,i} + c \cdot L_{rank3,i}\right) \times \left[d \cdot R_{urbanarea,i} + e \cdot \left(1 - R_{urbanarea,i}\right)\right] \tag{4}$$

where, $i$ represents grid ID, $p$ represents province, $n$ is the grid number in each province; $E_p$ is the total vehicular NH$_3$ emissions
of province $p$ in 2019, units in t; $E_{p,i}$ is the vehicular NH$_3$ emissions allocated to grid $i$, units in t; $R_{traffic,i}$ is the traffic indicator
in grid $i$, defined by road length of different road types ($L_{rank,i}$, rank1-3 represents for highway, arterial road and residential
road) and the urban area ratio ($R_{urbanarea,i}$); $a$-$e$ are allocating factors, referring to the traffic flow ratio of different road types in
urban and rural areas in Beijing (Yang et al., 2019). Here, $a$-$e$ are 1, 0.4, 0.3, 0.8, and 0.2, respectively. The digital road map
was obtained from the latest OpenStreetMap data for China (Openstreetmap, 2022). Urban area ratio was calculated based on
the urban residential areas within each grid.

We compared the gridded allocation results of on-road NH$_3$ emissions with the estimations based on link-level inventories in
four megacities in China (i.e., Beijing (Yang et al., 2019), Shanghai (An et al., 2021), Shenzhen (Wen et al., 2020) and Chengdu
(Wen et al., 2022b)), shown in Fig S1. Link-level NH$_3$ emission inventories were calculated based on the traffic profiles of the
whole road network obtained in our previous studies (Wen et al., 2022a; Wen et al., 2020; Yang et al., 2019) and NH$_3$ EFs
derived in 2.1. The coefficient of determination (R$^2$) varied from 0.63 to 0.80 among four megacities, demonstrating the
accuracy of the allocation method for on-road emissions. We also compared the monthly variations of  total anthropogenic
NH$_3$ emissions derived in this study with surface NH$_3$ observations obtained from the Ammonia Monitoring Network in
different regions of China in Kong et al (Kong et al., 2019), shown in Fig S2. The monthly variations consistent well with NH$_3$
observations over different regions in China, demonstrating the accuracy of both spatial and monthly allocations.

## 3 Results and Discussion

### 3.1 Historical trend of vehicular NH$_3$ emissions in China

NH$_3$ EFs of LDGVs and HDDVs derived in this study are compared with relative literatures conducted by dynamometer,
remote sensing, PEMS and other field measurements, as shown in Fig 1 (details listed in Table S2 and S3). NH$_3$ EFs of LDGVs
decreased significantly with the upgrading of vehicle emission standards, consistent with trends in other literatures. NH$_3$ EFs
of HDDVs without SCR are negligible (4.4 ± 2.4 mg/km), while the introduction of SCR systems greatly increased the risks
of ammonia slip (73.9 ± 118.7 mg/km). The introduction of Ammonia Slip Catalyst (ASC) in heavy-duty China VI emission





standard would significantly reduce NH$_3$ emissions of HDDVs (Mendoza-Villafuerte et al., 2017). Since China VI HDDVs have not been widely deployed until 2020, we didn't take SCR+ ASC into consideration in the calculation of NH$_3$ emission
inventories.

Taking the phase in of emission standards into consideration, the trends of annual and fleet average NH$_3$ EFs for gasoline and diesel vehicles in China from 2000 to 2019 are shown in Fig S3. NH$_3$ EFs for gasoline vehicles decreased from 66.6 mg/km in 2000 to 16.0 mg/km in 2019 due to the continuously upgrading of emission standards. NH$_3$ EFs of diesel fleet were negligible before 2014, while started to increase with the national implementation of China IV in 2014. Fleet average NH$_3$ EF of diesel
vehicles has surpassed gasoline vehicles in 2016 and increased to 36.5 mg/km in 2019.

The annual vehicular NH$_3$ emissions increased from 32.8 kt/yr to 87.1 kt/yr from 2000 to 2019 in China, as shown in Fig 2. The continuously increase from 2000 to 2010 mainly resulted from the rapid growth of gasoline vehicle ownership. However, emissions from gasoline vehicles started to decrease with the upgrading of emission standards in the past decade. NH$_3$ emissions of gasoline vehicles estimated based on bottom-up method agreed well with the top-down estimations based on
annual gasoline consumption and fuel-based EFs (see Fig S4). Emissions from diesel vehicles grew significantly under the joint effects of increasing HDDV populations and the rapid introduction of SCR systems since 2014. The emission proportion of diesel fleet grew significantly from less than 3% before 2014 to 33% in 2019. With the implementation of heavy-duty China VI emission standard in 2020, NH$_3$ emissions from diesel vehicles might be probably controlled. Gasoline vehicles (mainly LDGVs) will keep dominating the total on-road NH$_3$ emissions in the near future.

**3.2 Spatial and temporal distributions of on-road NH$_3$ emissions in China**

Highly spatial-resolved (3 km×3 km) vehicular NH$_3$ emission intensities in China in 2019 are illustrated in Fig 3. On-road emissions are distributed along road network, and the emission hotpots are highly correlated with densely-populated areas, which is different from the spatial distribution of agricultural emissions (see Fig S5). On-road NH$_3$ emission in two of the most populous regions, i.e., Beijing-Tianjin-Hebei (BTH) region, and the Yangtze River Delta (YRD) are also illustrated in Fig 3
(b) and (c). The average on-road NH$_3$ emission intensities in mainland China, BTH and YRD are 9.3, 42.4 and 46.5 kg·km$^{-2}$·yr$^{-1}$, with average population densities of 146, 511, and 668 person·km$^{-2}$ in 2019, respectively. It's important to note that on-road NH$_3$ emissions are positive correlated with population density, which will be further analyzed in 3.3.

We analyzed the contribution of on-road NH$_3$ emissions in total anthropogenic emissions in 2019. Agriculture (including livestock and fertilizer) dominated the total anthropogenic NH$_3$ nationwide (>90%), and the contribution of vehicular NH$_3$
emissions is insignificant comparing with agricultural emissions (<1%). However, the proportion of vehicular emissions varied significantly (from 0.36% to 8.91%) among provinces (see Fig S6). Beijing and Shanghai are the top two provinces with the highest vehicular NH$_3$ emission contributions in China, which are 8.91% and 7.33%, respectively, comparing with the nationwide level of 0.95%. Beijing and Shanghai are not only the core cities for BTH and YRD regions, but also two of the most populous megacities in China (with residential populations over 20 million). Thus, we chose Beijing and Shanghai as
typical cities to discuss hereinafter.





Though several studies have pointed out the significant temperature-dependence of NH$_3$ emissions from LDGVs (Selleri et al., 2022; Suarez-Bertoa et al., 2017), few studies have considered the seasonal variation of on-road NH$_3$ emissions in either inventory or air quality modeling. In this study, the temperature impacts on NH$_3$ emissions have been depicted by the comprehensive EF model. The fleet average NH$_3$ EFs of gasoline vehicles in February were 1.50 and 1.41 times of those in

August for Beijing and Shanghai, respectively, consistent with the NH$_3$ emission ratio of 1.4~2.1 reported in dynamometer measurements conducted under -7 ℃ relative to 23 ℃ (Selleri et al., 2022; Suarez-Bertoa et al., 2017). As the monthly variations of agricultural emissions (higher in summer than winter) are opposite to vehicular emissions, the vehicle emission proportions are significantly higher in winter. As shown in Fig S7, the city-scale vehicular NH$_3$ emission proportions are up to 14% and 12% in winter in Beijing and Shanghai, respectively, nearly twice of the annual average ratio of 8.9% and 7.3%.

The proportion would be even larger in urban areas, posing substantial risks for haze episodes during the wintertime.

### 3.3 Relative contribution of on-road and agricultural NH$_3$ emissions among different population density

The highly spatial-resolved NH$_3$ emission inventory enables us to distinguish the relative contribution of vehicular and agricultural emissions among various population density. Population density data in mainland China were extracted from WorldPop (Tatem, 2017) at a spatial resolution of 100 m (see Fig S8), then aggregated into 3 km to match the resolution of

emission inventories. As shown in Fig 4, the distribution of on-road NH$_3$ emission intensity is positive correlated with population density, while the trend in agricultural emission is opposite. NH$_3$ emission intensities of on-road traffic are much lower than agriculture for less populated areas, but the median will surpass agricultural sources in grids with population density higher than 10 thousand person/km$^2$. For extreme populous grids (population density>20 thousand person/km$^2$), agricultural emissions are less important comparing with on-road traffic emissions.

According to the statistics based on gridded emission inventories and population density, on-road NH$_3$ emissions exceed agricultural emissions in grids containing 23.0% of the Chinese population in 2019 (approximately 326.6 million people), and this number is up to 29.4% in winter. For densely-populated areas with population density higher than 2000 person/km$^2$, on-road NH$_3$ emissions exceed agricultural emissions where 53.3% of the population resides (approximately 287.8 million people), and up to 66.2% in winter. As two of the most populous megacities in the world, Beijing and Shanghai has 21.9 and 24.8

million residents in 2019. As shown in Fig 5, on-road NH$_3$ emissions in Beijing and Shanghai are significantly concentrated in densely-populated areas where agricultural emissions seldom exist, and gasoline vehicles accounted for most of these emissions due to the strict restrictions of heavy-duty trucks in central urban areas. The statistics show that on-road NH$_3$ emissions exceed agricultural emissions where 69.2% and 72.0% of population resides in Beijing and Shanghai, respectively. Thus, the significant role of on-road NH$_3$ emissions in populated areas and in winter may have been underappreciated without

taking into account the temporal and spatial variations of on-road emission inventories. As another important reactive nitrogen species besides NO$_X$, the significance of NH$_3$ emission control has not been fully addressed. Serving as a major contributor to both NO$_X$ and NH$_3$ emissions in urban areas, multipollutant control strategy for vehicular NO$_X$ and NH$_3$ emissions may be a more effective pathway to mitigate PM$_{2.5}$ pollution in densely-populated areas.



## 4 Conclusions

In this study, we established a comprehensive vehicular $NH_3$ emission factor model and compiled a gridded on-road $NH_3$ emission inventory with high spatial (3 km × 3 km), and temporal (monthly) resolutions for mainland China. $NH_3$ EFs for gasoline vehicles decreased from 66.6 mg/km in 2000 to 16.0 mg/km in 2019 due to the continuously upgrading of emission standards. The annual vehicular $NH_3$ emissions increased from 32.8 kt/yr to 87.1 kt/yr from 2000 to 2019 in China, mainly resulted from the rapid growth of gasoline vehicle ownership. On-road $NH_3$ emissions are significantly concentrated in

densely-populated areas where agricultural emissions seldom exist. It is found that on-road $NH_3$ emissions exceed agricultural emissions in grids containing 23.0% of the Chinese population in 2019 (approximately 326.6 million people), and this ratio is up to 29.4% in winter. For extreme populous cities such as Beijing and Shanghai, on-road $NH_3$ emissions exceed agricultural emissions where 69.2% and 72.0% of population resides, respectively.

This study gave a better insight into the absolute value and relative importance of on-road $NH_3$ emissions in different regions,

seasons and population densities in China, which is important in terms of the air quality implications. We emphasized the significant role of on-road $NH_3$ emissions in populated urban areas which may be underappreciated previously, highlighting the necessity to consider vehicular $NH_3$ emissions in air quality simulations. In addition, we clearly depicted the seasonal variation of on-road $NH_3$ emissions in our inventory by considering temperature-dependence of $NH_3$ emissions from LDGVs. As the monthly variations of agricultural emissions (higher in summer than winter) are opposite to vehicular emissions (higher

in winter than summer), the city-scale vehicular $NH_3$ emission proportions in winter are nearly twice of the annual average ratio in Beijing and Shanghai. This finding reminds us of the possibly severer risks for haze episodes during the wintertime. Precise air quality simulations based on the highly-resolved $NH_3$ emission inventory are required to quantify the relative contribution of on-road $NH_3$ emissions to urban $PM_{2.5}$ pollution in different seasons.

Although several pathways for agricultural emission abatement have been raised (Sha et al., 2021), the regulation for

agricultural $NH_3$ emissions is difficult due to both the technical gap and potential obstruction from stakeholders (Plautz, 2018). However, mitigating vehicular $NH_3$ emissions could be more feasible compared with the control of agricultural emissions. With the implementation of heavy-duty Euro/China VI emission standard, $NH_3$ emissions from diesel vehicles are expected to be well-controlled (see Fig 1). The $NH_3$ emission problems from petrol vehicles have also been addressed by the coming Euro 7/VII standard (European Commission, 2022), and there are proven aftertreatments to ensure Euro 7/VII vehicles to comply

with these proposed limits (Torp et al., 2021). Except for regulations from emission standards, traffic management for passenger vehicle fleet and promotion of electric vehicles can also significantly mitigate vehicular $NH_3$ emissions in urban areas. Therefore, the control of on-road $NH_3$ emissions can be a feasible and cost-effective way for mitigating haze pollution in urban areas, calling for great priority to strengthen regulations for vehicular $NH_3$ emissions worldwide.



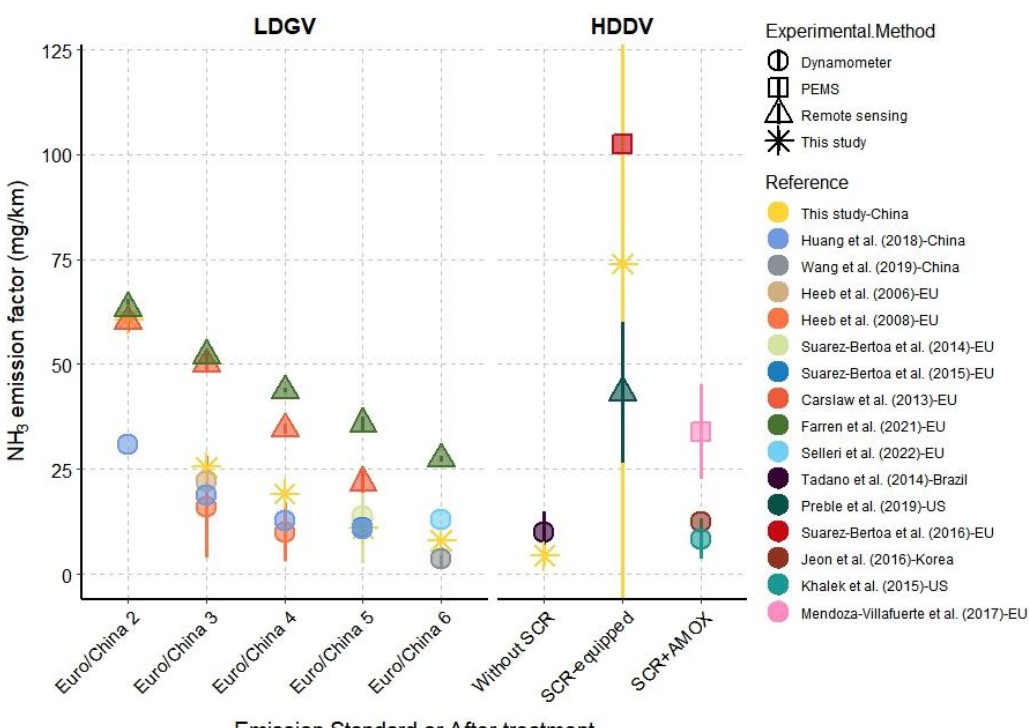

**Figure 1: Comparison of distance-based NH₃ EFs for of LDGVs and HDDVs in this study and other relative studies, disaggregated by emission standard or after-treatment technology.**

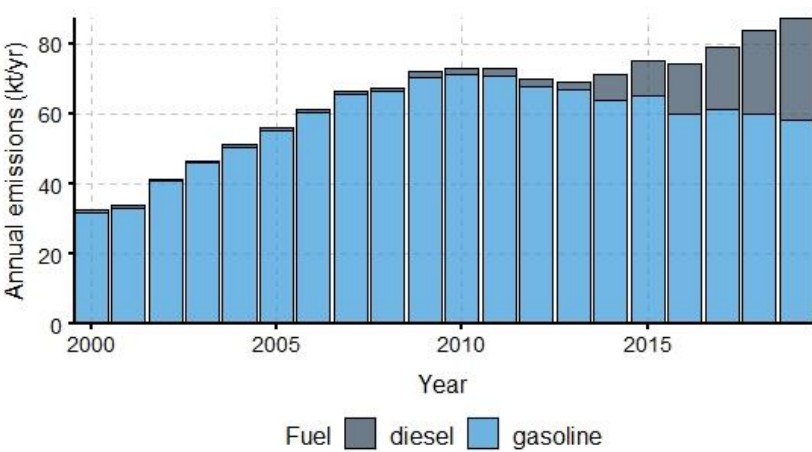

**Figure 2: Annual vehicular NH₃ emissions by fuel type in China, 2000-2019.**


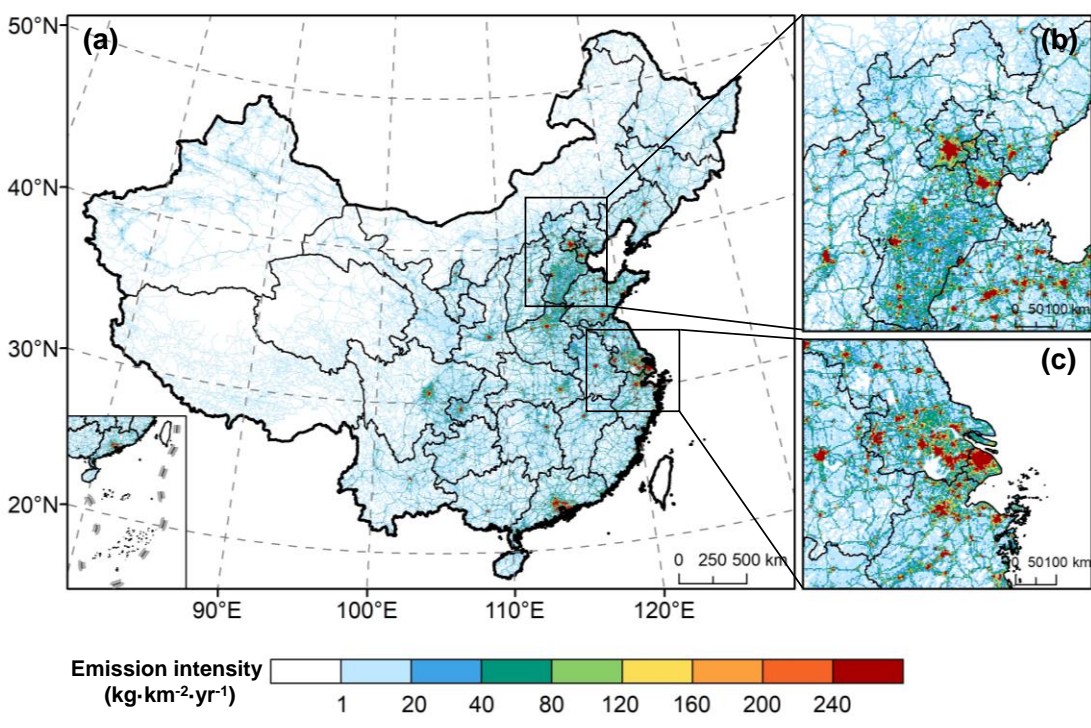

**Figure 3: Spatial distribution of on-road NH₃ emission intensities in (a) mainland China, (b) the Beijing-Tianjin-Hebei (BTH) region, and (c) the Yangtze River Delta (YRD) in 2019.**

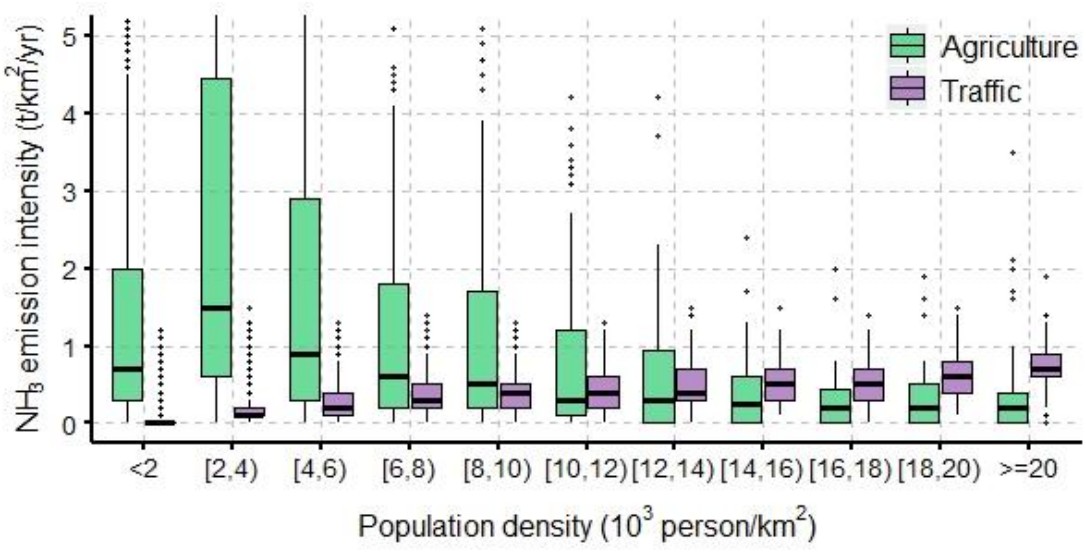

**Figure 4: Distribution of agricultural and on-road NH₃ emissions among different population densities in 2019.**





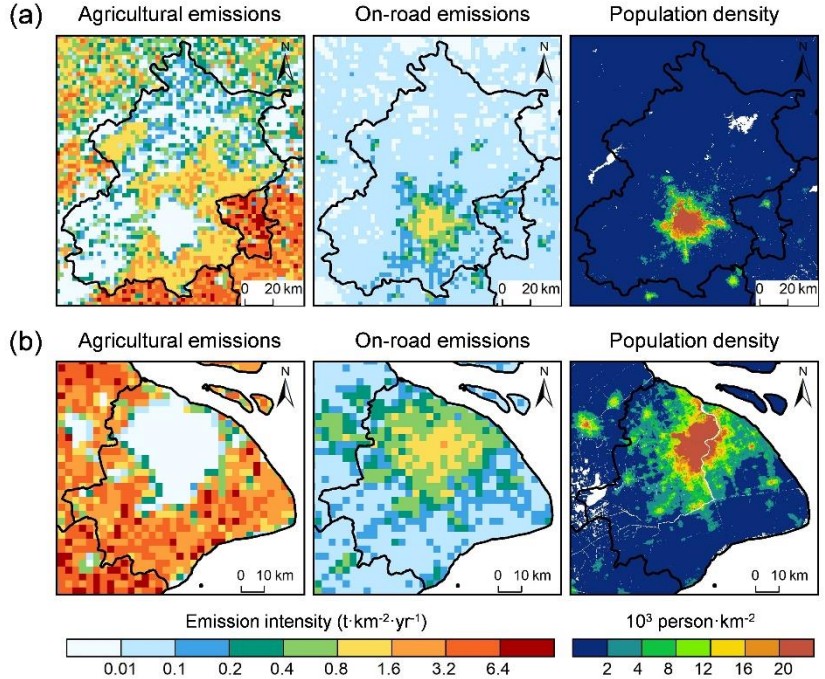

**Figure 5: Spatial distributions of agricultural and on-road NH₃ emission intensities, and population density in (a) Beijing and (b) Shanghai in 2019.**

**Author Contributions**

**Yifan Wen**: Data curation, Methodology, Visualization, Writing- Original draft preparation.

**Shaojun Zhang**: Conceptualization, Writing – review & editing, Supervision.

**Ye Wu**: Supervision.

**Jiming Hao**: Supervision.

**Competing interests**

The authors declare that they have no conflict of interest.



**Acknowledgments**

We are grateful to the National Key Research and Development Program of China (Grant No. 2022YFC3703600), the China National Postdoctoral Program for Innovative Talents (No. BX20220179) and the Shuimu Tsinghua Scholar Program (No. 2022SM011). All the authors are grateful to the Elsevier Language Editing Services for polishing the English expressions in this paper.

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
