# Peer review of "Vehicular ammonia emissions: An underappreciated emission source in densely-populated areas"

_Atmospheric Chemistry and Physics, 2022_

## Author Comment (AC1)

Responses to Anonymous Referee #1

**General comments**

In this work, the authors established a comprehensive vehicular NH3 emission model and compiled a gridded on-road NH3 emission inventory with high spatial (3 km × 3 km), and temporal (monthly) resolutions for mainland China using published NH3 emission factors of motor vehicles and their relevant impact factors. With this high-resolution emission inventory, vehicular NH3 emissions during the period of 2000-2019 were estimated. The authors showed that vehicular NH3 emissions could exceed agricultural emissions in the densely populated areas, especially for the extreme populous megacities such as Beijing and Shanghai. Although this conclusion is not unexpected, the paper gives a quantifiable and reliable result, which is valuable for future study. The paper is overall well written. I still have some doubts about the uncertainties of the vehicular NH3 emission inventory and some minor questions as listed below. I recommend publication after these issues are addressed.

**Specific comments**

1. Line 100: "Bottom-up estimation of long-term vehicular NH3 emissions". My major concern here is whether the authors have considered the additional impacts of the enhancement of driving conditions caused by traffic congestion in densely populated areas on vehicular NH3 emission factors? Or maybe the authors can discuss some uncertainties caused by this factor in the consequent sections of the text. After all, this paper focuses on the importance of NH3 emissions from motor vehicles in densely populated areas.

We have established a speed correction module in $NH_3$ EF model to justify the discrepancy between real-world $NH_3$ EFs and the basic driving condition according to the correlations between $NH_3$ emissions and VSP (Huang et al., 2018) (see the figure below for speed corrections for LDGVs). Hence, it's highly possible to quantify the impacts of various driving conditions such as traffic congestion on vehicular $NH_3$ emissions if real-world speed monitoring data are available. As shown in Fig R1, the $NH_3$ EFs of LDGVs under traffic congestion (with average speed ~15 km/h) is about 30% higher than basic driving conditions (with average speed 25~30 km/h). $NH_3$ EFs also increase significantly under aggressive highway driving cycles (> 90 km/h).

[Figure]

**Fig. R1** Speed correction curve for NH$_3$ EF of LDGVs with average speed from 5 to 120 km/h relative to the basic driving condition (25~30 km/h).

However, the national NH$_3$ emission inventory in this study was established based on provincial-level statistical data but not link-level traffic profiles due to the lack of detailed traffic monitoring data in national wide. Thus, the EFs used in this study are those under basic driving conditions. To address the possible impacts of driving conditions on vehicular NH$_3$ emissions, we have added a discussion in the manuscript (shown as below).

Line 256-261: Note that the impacts of driving condition were not included in this study. It's well documented that LDGV NH$_3$ emissions are strongly dependent on driving conditions (Huang et al., 2018). Higher LDGV NH$_3$ emissions are found under both low-speed (Farren et al., 2021) and aggressive highway driving cycles (Huang et al., 2018). For urban areas easily affected by traffic congestion, vehicular NH$_3$ emissions can be further enhanced. It's important to quantify the impacts of traffic conditions such as traffic congestion on vehicular NH$_3$ emissions in urban areas if real-world speed monitoring data is available in future researches.

2. Line 190: I suggest that the temporal distributions of vehicular NH3 emissions can be moved to the main text, because this topic is one of the novelties of this study. It would be better if the authors could provide a set of temperature-depended NH3 emission factor correction factors for reference.

We have introduced the ambient temperature corrections for NH$_3$ EFs in section 2.1 in the manuscript. To better address the novelty about temporal distributions of vehicular NH$_3$ emissions, the figure below illustrates the monthly variations in NH$_3$ EFs for LDGVs in Beijing and Shanghai generated from the NH$_3$ EF model. We have added this plot in the manuscript (Fig 4).

[Figure]

**Fig. R2** Monthly NH$_3$ EFs for LDGVs of various emission standards in (a) Beijing, and (b) Shanghai in 2019.

3. Line 196: If possible, I suggest that the authors could add more discussions on NH3 emissions from residential sources. According to Figure S7, Beijing and Shanghai also have a relatively high proportion of NH3 emissions from residential sources. Their emissions in each season are even higher than those from motor vehicles, and their emissions, if not unexpected, should also be mainly concentrated in densely populated areas.

NH$_3$ emission data from other anthropogenic sources used in this study were obtained from the updating works of Zheng et al (Zheng et al., 2019). Residential NH$_3$ emissions mainly include human excrement and domestic fuel combustion, and are mostly attributed to human activities in rural residential areas for megacities like Beijing and Shanghai (Zheng et al., 2019). Thus, even with high emission contributions in the whole city, residential emissions may not be as influential as traffic emissions in urban areas. We have added this discussion in the manuscript (line 218-222).

4. Line 219: Due to the lack of measurements of vehicular NH3 emission factors, NH3 emission inventory still has large uncertainty on the whole. Especially for diesel vehicles, abnormal urea uses and different SCR control strategies will affect its NH3 emission factor. Therefore, it is suggested that the authors could discuss more on the uncertainty of vehicular NH3 emission inventory.

We referred to the error bars of NH$_3$ emission measurements from various studies (Table S2 and S3) to estimate the uncertainty ranges of gasoline and diesel vehicles under different emission standards, shown as below. For diesel vehicles, the large uncertainty ranges in SCR-equipped EFs have involved the impacts of abnormal urea uses and different SCR control strategies.

**Table R1.** Uncertainty ranges of NH$_3$ emission factors.

| Vehicle types | Emission standards | Uncertainty ranges |
| --- | --- | --- |
| LDGV | Euro/China 2 | 4% |
| LDGV | Euro/China 3 | 27% |
| LDGV | Euro/China 4 | 25% |
| LDGV | Euro/China 5 | 33% |
| LDGV | Euro/China 6 | 38% |
| HDDV | Without SCR | 52% |
| HDDV | SCR-equipped | 81% |
| HDDV | SCR+AMOX | 45% |

Trends of fleet average NH$_3$ EFs for gasoline and diesel vehicles with uncertainty ranges are show in Fig R3. We have replaced Fig S3 with the figure below.

[Figure]

**Fig.R3** Trends of fleet average $NH_3$ EFs for gasoline and diesel vehicles in China, 2000-2019. Shadows show the uncertainty ranges.

We calculated the corresponding uncertainty in total emissions based on the uncertainty ranges in emission factors, shown as below. We have added discussions about uncertainty of vehicular $NH_3$ emission inventory in the manuscript (Line 163-164) and replaced Fig 2 with the figure below.

[Figure]

**Fig. R4** Annual vehicular $NH_3$ emissions by fuel type in China with uncertainty ranges, 2000-2019.

References:

Farren, N. J., Davison, J., Rose, R. A., Wagner, R. L., and Carslaw, D. C.: Characterisation of ammonia emissions from gasoline and gasoline hybrid passenger cars, Atmospheric Environment: X, 11, 10.1016/j.aeaoa.2021.100117, 2021.

Huang, C., Hu, Q., Lou, S., Tian, J., Wang, R., Xu, C., An, J., Ren, H., Ma, D., Quan, Y., Zhang, Y., and Li, L.: Ammonia Emission Measurements for Light-Duty Gasoline Vehicles in China and

Implications for Emission Modeling, Environmental Science & Technology, 52, 11223-11231, 10.1021/acs.est.8b03984, 2018.

Streets, D., Bond, T., Carmichael, G., Fernandes, S., Fu, Q., He, D., Klimont, Z., Nelson, S., Tsai, N. Y., Wang, M., Woo, J., and Yarber, K.: An inventory of gaseous and primary aerosol emissions in Asia in the year 2000, J Geophys Res-Atmos, 108, 10.1029/2002jd003093, 2003.

Zheng, H., Zhao, B., Wang, S., Wang, T., Ding, D., Chang, X., Liu, K., Xing, J., Dong, Z., Aunan, K., Liu, T., Wu, X., Zhang, S., and Wu, Y.: Transition in source contributions of PM2.5 exposure and associated premature mortality in China during 2005–2015, Environment International, 132, 105111, https://doi.org/10.1016/j.envint.2019.105111, 2019.

---

## Author Comment (AC2)

**Responses to Anonymous Referee #2**

During the past years, PM$_{2.5}$ pollution have been reduced substantially, while the occasionally occurred heavy PM$_{2.5}$ episodes and its driving forces still need to be explored. One of non-common sense hotspots is the role of NH$_3$. The relative importance of traffic sources to NH$_3$ emissions is still under debate.

Base on emission measurement data throughout different cities in China, this study developed high-quality traffic emission inventory of vehicular NH$_3$ emissions. This work can give a better insight into the absolute value and relative importance of vehicular NH$_3$ emissions in different regions, seasons and population densities in China. According to the results, they show that the significant role of on-road NH$_3$ emissions in populated areas have been underappreciated, which is quite important in terms of the atmospheric chemistry and air quality implications.

Overall, this manuscript is well organized and presented with some new insights on NH3 emissions and its contribution, thus, I think this paper is suitable for publishing in ACP after well addressing the following comments, questions and suggestions.

1.  As shown in Fig. 2, NH$_3$ emissions from gasoline vehicles have already declined since 2010 while emissions from diesel vehicles grew significantly since 2014. What's the most possible trend in total on-road NH$_3$ emissions in the near future under the join effects of vehicle growth and turnover?

The figure below shows the possible trend in total on-road NH$_3$ emissions in the near future under the join effects of vehicle growth and fleet turnover (impacts of COVID19 are not considered). Evolution of China's vehicle fleet in future is predicted based on the methodology in Wu et al (Wu et al., 2017). Total vehicular NH$_3$ emissions will reach the peak around 2020. NH$_3$ emissions from gasoline vehicles will keep decreasing in the next 5 years, while those from diesel vehicle also start to decrease with the implements of China VI emission standard since Jul 2021.

[Figure]

**Fig R1.** Annual vehicular NH$_3$ emissions by fuel type in China, 2000-2025.

2. I witnessed an overall higher level of EFs estimated from remote sensing than dynamometer measurements in Fig. 1. What's the possible reason? Some discussion should be added.

Dynamometer measurements are conducted for limited vehicles under the type-approval test cycles with controlled environmental condition. While remote sensing measurements always involve a huge number of vehicles and a complex mix of factors affecting vehicle emissions, such as fleet composition, driving conditions, environmental impacts, etc. As shown in Fig R1 and R2 of my reply to referee comment 1, $NH_3$ emissions are highly affected by ambient temperature and driving speed. Most of the remote sensing references mentioned in this paper contain measurements under cold temperature and highway driving modes. Also, remote sensing is easily affected by air turbulences that mix the plume emissions of various vehicle categories. Thus, the overall higher level of EFs estimated from remote sensing may result from the impacts of various ambient environment and driving conditions, as well as interferences from high-emitters among the fleets or other vehicle types.

We have added this discussion in the manuscript (line 150-153).

3. Why the shares of vehicular $NH_3$ emissions in total anthropogenic $NH_3$ emissions show a large difference between the urban areas Shanghai and Beijing? More detailed discussion is required to quantitatively address the uncertainty (e.g., $NH_3$ emissions from residential or industrial sectors).

$NH_3$ emission data from other anthropogenic sources used in this study were obtained from the updating works of Zheng et al (Zheng et al., 2019). The shares of vehicular $NH_3$ emissions in total anthropogenic $NH_3$ emissions in Beijing and Shanghai are 8.91% and 7.33% in 2019, respectively. As shown in the figure below, $NH_3$ emissions from residential and traffic sector are similar in Beijing and Shanghai, while industrial emissions are neglectable. The higher share of vehicular $NH_3$ emissions in Beijing is mainly result from lower agricultural emission share (i.e., livestock productions and fertilizer applications). The differences in agricultural emissions result from the differences in ambient temperature, type of fertilizer, the application rate of fertilizer and number of breeding stock in two cities.

Uncertainties in $NH_3$ emissions from other sectors may be brought by activity data and emission factors. Activity data from other emission sources were obtained from authoritative statistics in China to minimize the uncertainties (Zheng et al., 2019). Based on the methodology of uncertainty analysis from Dong et al (Dong et al., 2010) and Fu et al (Fu et al., 2015), the uncertainty ranges of industrial and residential emissions are -83%~127% and -58%~66%, respectively.

[Figure]

**Fig R2.** NH$_3$ emission contributions by sources in Beijing, and Shanghai in 2019.

4. This paper mainly focused on emission inventory but didn't reach to the real impacts on air quality. I understand that the scope of this paper may not be stretched further, but still wonder whether there's any evidence for the air quality impacts of on-road NH3 emissions in urban areas?

In another paper that has been accepted by EST recently, we conducted air quality simulations in Beijing and Shanghai, China, to access the impacts of on-road NH$_3$ emissions on air quality in urban areas. The results show that local vehicular NH$_3$ emissions could be responsible for approximately 3% of urban PM$_{2.5}$ concentrations during wintertime, and the contributions could be much higher during polluted periods (~3 μg/m$^3$) due to the high transformation ratio from the gaseous NH$_3$ to particle-phase ammonium.

5. The significance, shortage and implications of this study is suggested to be added in the Conclusions sections.

We have added discussions about the major limitation of this study in the Conclusions section, that we didn't account for the impacts of driving conditions on vehicular NH$_3$ emissions due to the lack of detailed traffic monitoring data in national wide (Line 256-261).

Reference:
Dong, W., Xing, J., and Wang, S.: Temporal and spatial distribution of anthropogenic ammonia emissions in China: 1994-2006, Huan jing ke xue= Huanjing kexue / [bian ji, Zhongguo ke xue yuan huan jing ke xue wei yuan hui "Huan jing ke xue" bian ji wei yuan hui.], 31, 1457-1463, 2010.
Fu, X., Wang, S., Ran, L., Pleim, J., Cooter, E., Bash, J., Benson, V., and Hao, J.: Estimating NH3 emissions from agricultural fertilizer application in China using the bi-directional CMAQ model coupled to an agro-ecosystem model, Atmospheric Chemistry and Physics Discussions, 15, 745-778, 10.5194/acpd-15-745-2015, 2015.
Wu, Y., Zhang, S. J., Hao, J. M., Liu, H., Wu, X., Hu, J., Walsh, M. P., Wallington, T. J., Zhang, K. M., and Stevanovic, S.: On-road vehicle emissions and their control in China: A review and outlook, Science of the Total Environment, 574, 332-349, 10.1016/j.scitotenv.2016.09.040, 2017.
Zheng, H., Zhao, B., Wang, S., Wang, T., Ding, D., Chang, X., Liu, K., Xing, J., Dong, Z., Aunan, K., Liu, T., Wu, X., Zhang, S., and Wu, Y.: Transition in source contributions of PM2.5 exposure

and associated premature mortality in China during 2005–2015, Environment International, 132, 105111, https://doi.org/10.1016/j.envint.2019.105111, 2019.

---

## Author Comment (AC3)

**Responses to Anonymous Referee #3**

This study provides a comprehensive vehicular NH3 emission model with useful insight into spatial and temporal variations of vehicular NH3. The important role of NH3 emissions from vehicles in urban areas with higher population densities is highlighted, which could have important implications for PM2.5 and haze events. Overall the paper is well written and I recommend publication if the comments below can be addressed.

- Section 2.1. Please clarify how the NH3 emission factors were obtained. For gasoline vehicles, was NH3 measured directly or predicted based on correlation with MCE? Further information on sample sizes and whether the data represents a wide range of driving conditions is needed. What are the uncertainties associated with the NH3 emission factors?

For gasoline vehicles, $NH_3$ EFs were not measured directly, but predicted based on the correlation between $NH_3$ EFs and MCE. Original measurements of $NH_3$ emissions and the derivation of relationship between $NH_3$ and MCE (calculated based on CO and $CO_2$ EFs) are detailed in Huang et al (Huang et al., 2018). CO and $CO_2$ EFs under basic driving conditions were obtained from EMBEV model, the archetype model for China's National Emission Inventory Guidebook (Zhang et al., 2014). For diesel vehicles, $NH_3$ EFs were derived based on measurement data from a fleet of heavy-duty diesel vehicles (HDDVs) (China III to China V) using PEMS and dynamometer (He et al., 2020).

Having the EFs under basic driving conditions, we also established speed correction modules to justify the discrepancy between real-world $NH_3$ EFs and the basic driving condition. For gasoline vehicles, the speed correction curve was established according to the correlations between $NH_3$ emissions and VSP (Huang et al., 2018) (see Fig R1 for speed corrections for LDGVs). For diesel vehicles, the speed correction curves were fitted based on average $NH_3$ EFs tested under different driving conditions. Hence, it's highly possible to quantify the impacts of various driving conditions such as traffic congestion on vehicular $NH_3$ emissions if real-world speed monitoring data are available. However, the national $NH_3$ emission inventory in this study was established based on provincial-level statistical data but not link-level traffic profiles due to the lack of detailed traffic monitoring data in national wide. Thus, the EFs used in this study are those under basic driving conditions. To address the possible impacts of driving conditions on vehicular $NH_3$ emissions, we have added a discussion in the manuscript (Line 256-261).

[Figure]

**Fig. R1** Speed correction curve for NH$_3$ EF of LDGVs with average speed from 5 to 120 km/h relative to the basic driving condition (25~30 km/h).

As for the uncertainties in NH$_3$ EFs, we referred to the error bars of NH$_3$ emission measurements from various studies (Table S2 and S3 in SI) to estimate the uncertainty ranges of gasoline and diesel vehicles under different emission standards, shown as below.

**Table R1.** Uncertainty ranges of NH$_3$ emission factors.

| Vehicle types | Emission standards | Uncertainty ranges |
|---|---|---|
| LDGV | Euro/China 2 | 4% |
| LDGV | Euro/China 3 | 27% |
| LDGV | Euro/China 4 | 25% |
| LDGV | Euro/China 5 | 33% |
| LDGV | Euro/China 6 | 38% |
| HDDV | Without SCR | 52% |
| HDDV | SCR-equipped | 81% |
| HDDV | SCR+AMOX | 45% |

- Line 94 - 95 explains that NH3 emission factors of other diesel vehicles were calculated based on the relative fuel consumptions compared with HDDVs. It would be useful to highlight any limitations of this approach. It is also stated that the NH3 emissions varied significantly among tested HDDVs. How did you account for this?

This study estimated EFs of other diesel vehicles based on the relative fuel consumptions compared with HDDVs due to the lack of measurement data. This approach has obvious limitations and can be improved if more measurement data are available. Nevertheless, HDDVs accounted for 89.8% of the total NH$_3$ emissions from diesel vehicles in 2019. Thus, the uncertainties brought by EFs of other diesel vehicles are limited.

We have added a discussion about the limitations of the estimation of EFs for other diesel vehicles (Line 261-264).

- Many findings e.g. total vehicular NH3 (32.8 kt to 87.1 kt NH3 from 2000-2019), proportions of NH3 in different provinces (e.g. 8.91%) will be affected by the uncertainties in the NH3 emission factors. Provide estimates of uncertainty associated with these statistics.

Based on the estimated uncertainties of NH$_3$ EFs (Fig R1), trends of fleet average NH$_3$ EFs for gasoline and diesel vehicles with uncertainty ranges are show in Fig R2. We have replaced Fig S3 with the figure below.

[Figure]

**Fig.R2** Trends of fleet average $NH_3$ EFs for gasoline and diesel vehicles in China, 2000-2019. Shadows show the uncertainty ranges.

We calculated the corresponding uncertainty in total emissions based on the uncertainty ranges in emission factors, shown as below. The annual vehicular $NH_3$ emissions increased from 32.8±1.7 kt/yr to 87.1±37.5 kt/yr from 2000 to 2019 in China. Proportions of vehicular $NH_3$ emission in Beijing and Shanghai are 8.91±3.83% and 7.33±3.15%, respectively. We have added uncertainty ranges in results in the manuscript (Line 163-164, 183-185) and replaced Fig 2 with the figure below.

[Figure]

**Fig. R3** Annual vehicular $NH_3$ emissions by fuel type in China with uncertainty ranges, 2000-2019.

- Does the compilation of gridded NH3 emission inventories account for any effects of different traffic conditions?

The impacts of traffic conditions were not considered in compilation of the gridded $NH_3$ emission inventory due to the lack of detailed traffic monitoring data in national wide. We have addressed this limitation in conclusion section (Line 256-261).

- Figure 1. The authors should refer to the SI, which explains how g/kg EFs have been converted to mg/km. It is useful to explain potential reasons for observed differences. For example, the derivations of mg/km emissions from remote sensing have not been adjusted

to account for different driving conditions / fuel consumption, whilst dynamometer measurements may be lower than on-road emissions. Farren 2020 (ES&T) could be useful for mg/km NH3 EFs.

A fleet-averaged rather than a time-specific fuel consumption (g/s) was used to convert the mg/kg EFs to mg/km, thus the derivations of mg/km EFs from remote sensing have not been adjusted to account for different driving conditions / fuel consumption. We have added this explanation in the manuscript (line 150-154).

- Section 3.1. The literature suggests NH3 emissions from gasoline vehicles can increase as vehicles deteriorate / vehicle mileage increases. Do the trends consider this effect, which may be particularly important in the future if gasoline car ownership is increasing? It would also be useful to state the proportion of the proposed increase in NH3 from diesel vehicles that can be attributed to HDDVs and therefore how this may change with implementation of China VI.

The deviations in $NH_3$ EFs of gasoline vehicles caused by deterioration were aggregated into various emission standards in our model framework. $NH_3$ EFs under a certain emission standard vary with different model years. Thus, the trends in Fig 1 have considered the effects of deterioration.

We have provided a prediction of $NH_3$ emission trends in the near future in response to RC2. The figure below shows the possible trend in total on-road $NH_3$ emissions in the near future under the join effects of vehicle growth and fleet turnover (impacts of COVID19 are not considered). Evolution of China's vehicle fleet in future is predicted based on the methodology in Wu et al (Wu et al., 2017). Total vehicular $NH_3$ emissions will reach the peak around 2020. $NH_3$ emissions from gasoline vehicles will keep decreasing in the next 5 years, while those from diesel vehicle also start to decrease with the implements of China VI emission standard since Jul 2021.

[Figure]

**Fig R4.** Annual vehicular $NH_3$ emissions by fuel type in China, 2000-2025.

- Conclusion. This study provides useful insight into vehicular NH3 emissions. It is recommended that the conclusions address the limitations of this study and how this could be improved in the future to better understand the air quality impacts of vehicular NH3.

We have added discussions about the major limitations of this study in the Conclusions section (Line 256-264). Firstly, impacts of driving condition were not included in this study. For urban areas with complex driving conditions and easily affected by traffic congestion, vehicular $NH_3$ emissions can be further enhanced. It's important to address the impacts of traffic conditions on vehicular $NH_3$ emissions in urban areas if real-world speed monitoring data is available in future works. Secondly, we estimated EFs of other diesel vehicles based on the relative fuel consumptions compared with HDDVs due to the lack of measurement data. This approach has obvious limitations and can be improved if more measurement data are available.

Technical corrections:

- Use of informal language e.g. line 41 'What's more', line 154, line 176.
- Line 144: 'The monthly variations compare well'
- Line 168: 'might be probably controlled' - be more specific
- Line 198: 'among various population densities.'
- Line 207: should this be 20,000 person/km2?

It's 2000 person/$km^2$ for sure. 2000 person/$km^2$ is higher than the population density of most of cities in China.

- Line 236: 'more severe'
- Line 244: 'Euro 7/VII vehicles comply

Technical corrections are modified accordingly.

Reference:

He, L. Q., Zhang, S. J., Hu, J. N., Li, Z., Zheng, X., Cao, Y., Xu, G., Yan, M., and Wu, Y.: On-road emission measurements of reactive nitrogen compounds from heavy-duty diesel trucks in China, Environmental Pollution, 262, 114280, 10.1016/j.envpol.2020.114280, 2020.

Huang, C., Hu, Q., Lou, S., Tian, J., Wang, R., Xu, C., An, J., Ren, H., Ma, D., Quan, Y., Zhang, Y., and Li, L.: Ammonia Emission Measurements for Light-Duty Gasoline Vehicles in China and Implications for Emission Modeling, Environmental Science & Technology, 52, 11223-11231, 10.1021/acs.est.8b03984, 2018.

Wu, Y., Zhang, S. J., Hao, J. M., Liu, H., Wu, X., Hu, J., Walsh, M. P., Wallington, T. J., Zhang, K. M., and Stevanovic, S.: On-road vehicle emissions and their control in China: A review and outlook, Science of the Total Environment, 574, 332-349, 10.1016/j.scitotenv.2016.09.040, 2017.

Zhang, S., Wu, Y., Wu, X., Li, M., Ge, Y., Liang, B., Xu, Y., Zhou, Y., Liu, H., Fu, L., and Hao, J.: Historic and future trends of vehicle emissions in Beijing, 1998–2020: A policy assessment for the most stringent vehicle emission control program in China, Atmospheric Environment, 89, 216-229, 10.1016/j.atmosenv.2013.12.002, 2014.

---

## Referee Report (RR1)

I support publication of the manuscript, subject to the following minor revisions:

- The uncertainties in annual vehicular NH3 emissions have been added to the manuscript (line 164). However these should also be included in the abstract.

- A detailed explanation of how the uncertainties have been derived (i.e. using measurements from various literature studies) should be provided. Table R1 (uncertainty ranges of NH3 emission factors) should be included in the manuscript.

- The emission uncertainties should be followed through to all statistics/results throughout the manuscript e.g. how does this potentially affect the proportion of grids where vehicular emissions exceed agricultural.

---

## Author Response (AR2)

**Responses to Editors and Reviewers**

**Public justification (visible to the public if the article is accepted and published)**:
This manuscript could be accepted after minor revisions following the reviewer's comments (see report #1) concerning the uncertainties and related issue.

● The uncertainties in annual vehicular NH3 emissions have been added to the manuscript (line 164). However these should also be included in the abstract.
We have added uncertainty ranges in the abstract.

● A detailed explanation of how the uncertainties have been derived (i.e. using measurements from various literature studies) should be provided. Table R1 (uncertainty ranges of NH3 emission factors) should be included in the manuscript.
We have explained the derivation of uncertainties in section 2.1 of the manuscript (Line 96-99) and added Table R1 into the Supplementary Information (Table S4).

● The emission uncertainties should be followed through to all statistics/results throughout the manuscript e.g. how does this potentially affect the proportion of grids where vehicular emissions exceed agricultural.
We have carefully revised the manuscript and added uncertainty ranges for all statistics through the manuscript.